# The Influence of Childhood Trauma and Family Functioning on Internet Addiction in Adolescents: A Chain-Mediated Model Analysis

**DOI:** 10.3390/ijerph192013639

**Published:** 2022-10-20

**Authors:** Manji Hu, Lin Xu, Wei Zhu, Tingting Zhang, Qiang Wang, Zisheng Ai, Xudong Zhao

**Affiliations:** 1Shanghai Pudong New Area Mental Health Center, School of Medicine, Tongji University, Shanghai 200124, China; 2Shanghai Yangjing High School, Shanghai 200122, China; 3Education Institute of Yangpu District, Shanghai 200092, China; 4Department of Medical Statistics, School of Medicine, Tongji University, Shanghai 200092, China; 5Clinical Research Center for Mental Disorders, Chinese-German Institute of Mental Health, Shanghai Pudong New Area Mental Health Center, School of Medicine, Tongji University, Shanghai 200124, China

**Keywords:** childhood trauma, internet addiction, systematic family dynamics, family functioning, anxiety and depression

## Abstract

Objective: This study aimed to examine the prevalence of Internet addiction in adolescents, analyze the associations of childhood trauma, systematic family dynamics, and family functioning with Internet addiction, and investigate the mediating chain role of anxiety and depression in the relationship of childhood trauma and family functioning with adolescent Internet addiction. Methods: This was a cross-sectional study in which general sociodemographic data were obtained from 3357 adolescents in grades 6–12 who were assessed using psychometric instruments such as the Childhood Trauma Questionnaire, Young Internet Addiction Test, Systematic Family Dynamics Self-Rating Scale (SSFD), Family Functioning Assessment (FAD), Self-Rating Depression Scale (SDS), and Self-Rating Anxiety Scale (SAS). Results: (1) The prevalence of Internet addiction among adolescents was 26.09% (876/3357). The prevalence of childhood trauma was 54.96% (1845/3357), and the prevalence of Internet addiction was significantly different between adolescents who suffered childhood trauma and those who did not (χ^2^ = 96.801, ν = 1, *p* = 0.000). (2) Childhood trauma and various dimensions of systematic family dynamics had a significant negative and positive relationship with poor family functioning and anxiety or depression, respectively. (3) Childhood trauma was a positive predictor of Internet addiction through the chain-mediated effect of anxiety and depression, but there were no direct effects. Poor family functioning was a positive predictor of adolescent Internet addiction, and this positive prediction was augmented by the chain-mediated effect of anxiety and depression. Conclusions: Childhood trauma and poor family functioning or support predicted Internet addiction in adolescents, with anxiety and depression as mediators.

## 1. Introduction

With the rapid development of the Internet and intelligent electronic devices [1], Internet access has become a part of people’s lives. After the coronavirus disease pandemic 2019 (COVID-19) broke out in China, adolescents spent more time with electronic devices through home online learning. Furthermore, the Internet population is significantly younger [2,3]. The Internet penetration rate of minors reached 94.9% by the end of December 2021, significantly higher than that of the general population, which is 73.0% [3]. In Europe and the United States, adolescents’ Internet use time has similarly increased significantly [4]. Although the Internet has been enormously beneficial to adolescents, the associated damage it has caused cannot be ignored [5]. In clinical practice, parents and adolescents have reported a significant increase in uncontrolled Internet use, thus causing a significant increase in reports of parent–child conflict and adolescent emotional, behavioral, concentration, and learning efficiency problems. Because adolescents are still in a critical stage of mental and physical development, their cognitive function, critical thinking [4], motivation, and impulse control, especially self-control abilities, are immature. The earlier exposure to the Internet using, the more likely it is to affect the processes of transformation and maturation of brain structures, thus interfering with the development of mental dimensions such as anticipation of reward, emotional processing, decision making, and impulse control, leading to a higher risk of Internet Addiction (IA) and more serious psychological and behavioral problems for adolescents [4,6,7,8,9,10,11,12,13,14]. The prevalence of adolescent IA since the beginning of the COVID-19 pandemic was significantly higher than that observed before the pandemic [6,15,16,17,18].

Young and Rogers defined adolescents’ problems as early as 1996 [19], and researchers have found that IA causes significant damage to adolescents, including their mental health, physical health, and social functioning. IA also leads to emotional problems such as anxiety, depression, irritability [16], impaired behavioral control, impaired concentration and execution, personality changes, and negative self-perception [20,21,22]. However, adverse mental states exacerbate adolescent Internet use behavior [23]. IA can also disrupt sleep patterns and damage immature brain structure and function [21,22]. Eventually, IA among adolescents leads to many social function impairments, such as academic failure, interpersonal withdrawal, and further deterioration of interpersonal relationships [24].

The causes of adolescent IA are widespread and complex [2,25]. Among various intrapersonal factors, an individual’s childhood traumatic experience is the highest risk factor for adolescent IA [26]. Childhood trauma is a global phenomenon [27,28], and can cause serious psychological damage to an individual, leading to anxiety, depression, and other emotional and addictive behaviors [21,29,30]. Among external environmental factors, poor family environment and family dynamics may be important risk factors for IA [31,32]. McMaster’s model theory of family functioning states that the role of the family is to provide its members with the appropriate environmental conditions needed for physical, psychological, and social adaptations to growth [33]. Parents’ excessive psychological control over adolescents, poor communication, and rigid relationships with their children, as well as poor emotional relationships among family members, contribute to adolescents’ emotional and behavioral problems [6,11,32,34]. Family dysfunction is strongly associated with childhood trauma [35], which drives adolescents into the virtual world on multiple levels, leading to an increased risk of IA [6,36,37,38,39].

Based on the many causative factors associated with IA, this study aimed to identify the key causative factors and models that cause IA in adolescents to explore effective intervention methods, which will reduce the damage caused by IA and promote the healthy development of the mental, physical, and social functions of adolescents.

Therefore, this study proposed the following hypotheses related to IA.

Adolescents’ childhood trauma and dysfunctional family are high-risk causative factors of IA.Adolescents’ anxiety and depression are associated with childhood trauma and family dysfunction.Anxiety and depression mediate the relationship between childhood trauma and poor family functioning with IA.

## 2. Methods

### 2.1. Participants

This cross-sectional study was conducted in Shanghai, China, from October to December 2020. To estimate the sample size, 189 samples were pre-tested. The prevalence of IA (π) obtained from the pretest was approximately 23.4%; a 6% relative error was allowed in this study. The absolute error was calculated as δ = 0.06π = 0.06 × 23.4%. A 95% confidence interval (CI) was adopted, and *μ*_a_ was 1.96. The sample size was determined using the following equation: n = [1.96^2^ × 23.4% × (1–23.4%)]/(0.06 × 23.4%)^2^, which was ≈ 3493. Considering the possibility of invalid cases, the sample size must be increased by 15%; therefore, the final sample size was calculated as 3493/(1–15%), which was ≈ 4109. 

After determining the sample size, the multi-stage stratified whole-group sampling method, in which two districts were randomly selected among 17 communities and counties in Shanghai, three junior high schools and three high schools were chosen in each section, and two classes were randomly selected in each grade, was used. All students in the selected classes were surveyed with the consent of their guardians, and after both, the guardians and students signed informed consent forms. A total of 4109 questionnaires were distributed, and finally, excluding incomplete questionnaires, a total of 3357 (81.7%) valid questionnaires were retrieved.

Inclusion criteria: (1) adolescents who were enrolled in public schools at the junior high and high school levels at the time of the study and (2) adolescents who voluntarily participated in the psychological assessment survey and with their guardians signed an informed consent form.

Exclusion criteria: (1) adolescents who were enrolled in school at the time of the study but had suspended their studies for various reasons and (2) adolescents who were unwilling to participate in this psychological assessment survey.

### 2.2. Measures 

#### 2.2.1. Demographic Information Sheet 

A demographic information sheet, which collected information including sex, age, schooling grade, number of siblings, mode of living, parents’ generation, parents’ education level, parents’ marital quality, and self-assessment of family economic satisfaction (with a score of 0 to 10, 0 being little dissatisfaction and 10 being the maximal satisfaction) was distributed to the participants.

#### 2.2.2. Childhood Trauma Questionnaire-Short Form

The Childhood Trauma Questionnaire-Short Form (CTQ-28) is a retrospective self-report questionnaire developed by Bernstein et al. [8,40], which asks questions about traumatic experiences in early childhood and adolescence and is rated from 1 (never) to 5 (very often). For each type of trauma, scores ranged from 5 to 25, and for the total trauma score, scores ran from 25 to 125. The questionnaire assesses five types of childhood trauma, including emotional abuse (EA), physical abuse (PA), sexual abuse (SA), emotional neglect (EN), and physical neglect (PN). Three additional items were used for validity ratings. An individual who experienced childhood trauma was considered to have experienced EA if the dimension scores were ≥13, PA if scores were ≥10, SA if scores were ≥8, EN if scores were ≥15, and PN if scores were ≥10. Zhao et al. (2005) [40] translated and revised the Chinese version of the CTQ-28 scale and confirmed that it has been used in clinical studies successfully with good reliability, validity, and internal consistency and Cronbach α coefficient of 0.64 and re-measuring reliability of 0.75.

#### 2.2.3. Young-Internet Addiction Test 

A Chinese version of Young’s Internet Addiction Test (IAT-20) was used to measure IA [41,42]. In total, 20 items were rated on a 5-point scale, with one being “rarely” and five being “always,” and a final score was determined; the higher the total scores, the more prone an adolescent is to IA. The questionnaire is assessed based on the evaluation criteria of domestic researchers. A total score of <50 indicated no IA, and a total score of ≥50 indicated IA. The Cronbach’s α coefficient for this questionnaire was 0.90 [41,43].

#### 2.2.4. The Self-Rating Scale of Systematic Family Dynamics, Revised Version 

The Self-rating Scale of Systematic Family Dynamics, revised version (SSFD), was compiled by Zhao et al. [44], based on Heidelberg’s systematic family dynamics theory combined with the Chinese cultural background. The self-assessment questionnaire was revised and republished in 2014 [44]. This questionnaire includes the following four dimensions: family atmosphere (FA, eight items), individualized (IN, six items), systematic logic (SL, five items), and illness concept (IC, four items), totaling 23 items. Each item was scored on a 5-point scale: 1 = completely disagree; 2 = very much disagree; 3 = partially agree; 4 = very much agree; and 5= completely agree. There were both positive and negative items on the scale. The higher the positive item score, the more positive the item; the higher the negative item score, the less positive the item. Cronbach’s α and split-half correlation coefficients were 0. 79 and 0. 84, respectively [44].

#### 2.2.5. Family Assessment Device 

According to the McMaster family function model, the Family Assessment Device (FAD) scale includes seven subscales. This study used the general function subscale (GF) and behavior control subscale (BC). The GF consists of 12 self-report items, and the BC consists of 9 items. Each item is rated as strongly agree, agree, disagree, or completely disagree. The corresponding scores range from 1 to 4, and the Starred items were reversely scored. In this study, the higher the scale score, the less healthy the family function. Additionally, the FAD has demonstrated good reliability and validity in Chinese children over 12 years old [45]. The Cronbach’s α coefficient of BC is 0.71, and the correlation coefficient is greater than 0.5 [45].

#### 2.2.6. Self-Rating Anxiety Scale

The self-rating anxiety scale (SAS) compiled by Zung in 1971 reflects subjective feelings of anxiety, with a total of 20 items. The scores are summed to obtain a raw score, standardized to a cut-off of 50. Individuals with scores ≥50 were considered anxious, and those with scores <50 had no anxiety [45]. The Cronbach’s α and correlation coefficients of SAS were 0.697 and 0.777, respectively [46].

#### 2.2.7. Self-Rating Depression Scale 

The self-rating depression scale (SDS) was developed by Zung (1965) [45]. The questionnaire has 20 items that are scored using a 4-point Likert scale to evaluate depression. The sum of the raw scores ranges from 20 to 80. The cut-off after standardization is 53 points; therefore, scores more than or equal to 53 points indicate depression and those less than 53 points indicate no depression [45]. The Cronbach’s α and correlation coefficients of the SDS are r.73 and 0.84, respectively [45].

### 2.3. Statistical Analyses

Data were analyzed using IBM SPSS Statistics for Windows, version 25.0 (IBM Corp., Armonk, NY, USA). First, we coded the options for each categorical variable in Table 1. Second, Pearson’s chi-square test was used to analyze the differences in IA rates between the groups. Third, Pearson’s bivariate correlation analysis was used to analyze the correlation between IA and other variables. Finally, a regression analysis of the chain-mediation model was performed using PROCESS Macro for SPSS 3.3 [47], using Model 6 with 5000 bias-corrected bootstrap samples. Statistical significance was determined using a *p*-value of 0.05. Indirect effects were analyzed at a significance level of 0.05. If the bootstrap 95% confidence intervals (Cis) that do not include zero indicate significant effects at α = 0.05 [47].

### 2.4. Ethics

The present study was approved by the Ethical Committee of the Mental Health Center affiliated with Tongji University in Shanghai (No. PDJWLL2019008). Informed consent was obtained from all participants, including those younger than 18.

## 3. Results

### 3.1. Differences in the Prevalence of Internet Addiction among Adolescents according to Their Demographic Characteristics

The descriptive statistics of all participants with or without IA and their respective *p* values were presented in Table 1. The average age of 3357 young adults was 13.69 years (SD = 2.06). The prevalence of IA among adolescents was 26.2%. The following variables showed statistically significant intergroup differences in the prevalence of IA: whether the adolescent was a single-child or not, grades, parental preference for children, parental education level, parental marital quality, self-assessment of whether internet use is out of control, presence or absence of childhood trauma, and presence or absence of anxiety and depression. There were significant correlations between IA and the following variables: being a single child or not, schooling grade, parental preference of children, parental education level, parental marital quality, and satisfaction with family economic status, which were subsequently used as control variables for the regression analysis.

### 3.2. Internet Addiction in Adolescents with Childhood Trauma, Anxiety, or Depression

In this study, the prevalence of childhood trauma was 55.0% (59.1% of boys and 51.0% of girls), and the difference between the sexes was statistically significant (χ^2^ = 22.012, ν = 1, *p* = 0.000). However, the difference in IA prevalence was not statistically significant (χ^2^ = 0.105, ν = 1, *p* = 0.746). The prevalence of IA among adolescents with childhood trauma was 32.85%, significantly different from that among adolescents without childhood trauma (17.86%; χ^2^ = 96.801, ν = 1, *p* = 0.025 > 0.05). The prevalence of depression and anxiety in adolescents was 29.1% and 23.7%, respectively. The prevalence of IA in adolescents with childhood trauma according to the five types of traumas and negative emotions shown in Figure 1. The prevalence of IA or adverse emotions in adolescents with childhood trauma was significantly higher than the those without childhood trauma or emotional disorders.

### 3.3. Correlations between Internet Addiction in Adolescents and Other Variables 

Pearson’s correlation analysis showed a significant positive correlation between the IAT-20 scores and the SAS, SDS, the five CTQ-28 subscales, GF, and BC scores (Table 2). There was a significant negative correlation between the IAT-20 scores and all the SSFD dimension scores, except the IC scores. The correlation coefficients were more than 0.7 for the following variables: SAS and SDS, FA and IN or SSFD, IN and SSFD, and IC and IN or SSFD, 0.708, 0.703, 0.793, 0.825, and 0.720, respectively. 

### 3.4. Childhood Trauma and Poor Family Functioning as Predictors of IA

The proposed chain-mediated model (Model 6) was tested, with CTQ-28 total scores and GF scores as independent variables; IAT-20 scores as the dependent variable; anxiety and depression as mediating variables; and schooling grade, whether or not the adolescent is a single child, parental preference, parental education level, and parental marital quality as control variables. The results of the path coefficients are shown in Figure 2 and Figure 3. The CTQ-28 total scores were positive predictors of the SAS and SDS scores (Figure 2). The values with their bootstrap 95% CI are presented as following: CTQ-28 total scores as predictors of SAS scores: β = 5.3157, bootstrap 95% CI (4.2278, 6.4036); CTQ-28 total scores as predictors of SDS scores: β = 3.7308, bootstrap 95% CI (2.8393, 4.6222); GF as positive predictors of SAS: β = 0.6403, bootstrap 95% CI (0.5558, 0.7248) (Figure 3); and GF scores as positive predictors of SDS scores: β = 0.5052, bootstrap 95% CI (0.4324, 0.5780). All the bootstrap 95% CIs did not contain 0, verifying that Hypothesis 2 is true. The analysis of the direct effects of the CTQ-28 total scores on the IAT-20 scores was not significant (β = 1.1631, *p* > 0.05). However, the indirect effect of CTQ-28 total scores on IAT-20 scores was significant (β = 2.6155, *p* < 0.001), indicating a fully mediated effect, which verifies that Hypothesis 3 is true. The total effect of the CTQ-28 total scores on the IAT-20 scores was significant (β = 3.7786, *p* < 0.001). The direct effect of GF scores on the IAT-20 scores was significant (β = 0.2332, *p* = 0.0002 < 0.001), and the indirect effect via the SAS and SDS scores were also significant (β = 0.2789, *p* < 0.001). There was a partial mediating effect via the chain effect of the SAS and SDS scores, verifying that Hypothesis 3 is true. The total effect value of the GF scores on the IAT-20 scores was significant (β = 0.5121, *p* < 0.001), demonstrating that Hypothesis 1 is true.

The mediating effect of the CTQ-28 total scores on the IAT-20 scores via the SAS and SDS scores was significant (R^2^ = 0.1992, F [8, 1456] = 45.2615, *p* < 0.001). The results obtained after bootstrapping showed the indirect effect of anxiety and depression (β = 2.6155; 95% CI = [1.9555, 3.3105]; (Table 3), and there was no significant difference in the effect size between the three pathways.

The GF scores were positive predictors of the IAT-20 scores via the SAS and SDS scores, and the regression equation was significant (R^2^ = 0.2270, F [8, 1446] = 53.0771, *p* < 0.001). Furthermore, as shown in Table 4, our analysis indicates an indirect effect of anxiety and depression (β = 0.2789; 95% CI = [0.2068, 0.3544]). Among the three pathways, the indirect effect via the SAS scores was larger than that via the SAS–SDS scores (β = 0.1118, bootstrap 95% CI = [0.0231, 0.2089]); the CI value interval did not contain 0, and the effect difference was significant.

## 4. Discussion

### 4.1. The Variability of Internet Addiction Prevalence with Different Demographic Characteristics among Adolescents

The prevalence of IA among adolescents in the study was 26.2%, which is higher than that (16.5%) reported by Linyuan et al. (2021) [48]. Many current studies have shown that boys are prone to IA and have also said that the prevalence of IA among boys is higher than that among girls [2,35,49]. The differences in IA’s prevalence between genders were insignificant, possibly due to the different ways different genders use the Internet. The 49th China Internet Survey (2022) showed that underage netizens use the Internet for applications such as search engines, social networking sites, news and information, shopping, short videos, animation, and comics [3]. This study showed that the prevalence of adolescent IA was significantly associated with childhood trauma, anxiety and depression, and family dysfunction (poor parental marital quality, low family atmosphere, high family behavioral control, etc.). The poorer the parents’ marriage quality, the higher the prevalence of IA. Frequent and violent conflicts between parents are major factors that lead to family dysfunction, which can cause increased psychological stress or trauma to children in the family. Research has shown that parental marital conflict is a risk factor for adolescent IA [50,51,52]. the prevalence of IA in adolescents with childhood trauma was almost twice as high as the prevalence without childhood trauma. The prevalence of IA in adolescents with anxiety and depression was about 2.5 times higher than that without mood disorders. The worst the family functions, the higher the prevalence of IA in adolescents. Chen et al. (2020) [52] found that terrible family functions can lead children to become involved in the virtual world for temporary emotional support and a sense of belonging.

### 4.2. Effects of Childhood Trauma on Internet Addiction in Adolescents

According to the present study, boys were more likely to experience childhood trauma than that experienced by girls, and there was no significant difference in the prevalence of IA between the sexes. The prevalence of anxiety and depression was higher in girls than boys, with a substantial difference between the two sexes. Figure 1 shows the association of the prevalence of IA with different emotional disorders and traumatic experiences. The prevalence of IA with emotional abuse and anxiety was the highest. According to the chain-mediated effect results obtained using Process Macro Model 6 (Figure 2), childhood trauma has no direct positive predictive effect on IA but has a significant positive predictive effect on anxiety or depression. Anxiety and depression had significant positive predictive effects on IA. Moreover, childhood trauma and adverse emotions were positively associated with IA. As reported by other researchers, this study showed that anxiety and depression could increase the positive predictive effect of childhood trauma on IA [26,27,53]. Similar to many studies, childhood trauma is a risk factor for emotional dysregulation in adolescents, which further exacerbates the development of IA in adolescents [54,55,56].

### 4.3. Poor Family Functioning as a Predictor of Adolescent IA

According to the correlation analysis, the GF and BC scores were significantly correlated in instances where inappropriate control behaviors in the family could damage family functioning. GF scores were negatively correlated with each dimension of the SSFD scores, especially the FA scores. Higher FA scores indicated a more relaxed and pleasant FA, higher GF scores indicated unhealthy family functioning, and poor family functioning was associated with poor FA. The correlations between GF scores and SAS, SDS, and IAT-20 scores were significant, indicating that the less healthy the family functioning, the more likely adolescents are prone to anxiety, depression, and IA, as suggested by Ma (2021) [57] and Chen et al., (2020) [6] that poor family functioning is a risk factor for adolescent mood disorders. Through chain-mediated path analysis, poor family functioning showed a significant positive predictive effect on adolescents’ IA. In addition, poor family functioning is a risk predictor of adolescent emotional disorders. As suggested by Marzilli et al. (2020) [58], symptoms of negative emotions can directly increase the risk of IA, and anxiety and depression are strong risk predictors of IA. In this study, poor family functioning showed an increased risk-predictive effect on adolescents’ IA through the indirect mediated pathway of anxiety and depression. This result was also consistent with previous studies that have shown that a poor family environment is an important risk factor for IA [31,32,33,36,55].

### 4.4. Limitations and Directions for Future Research

There were some limitations as it is a cross-sectional survey in this study. First, although the questionnaires used in this survey had good reliability and validity, accuracy could not be ensured because the information was collected exclusively through self-reporting, and the researcher did not interview the participants. Second, experiences of childhood trauma, experiences of anxiety and depression, systematic family dynamics, and family functioning are subject to constant change; therefore, it is difficult to establish an accurate correlation between each of these variables and IA, despite accounting for control factors during the analysis. To improve the accuracy of the survey results, the primary guardians of the youth could be surveyed simultaneously in a future study. Finally, since we were not adjusting our analyses, there was also a chance for a residual confounding bias in this study. In the follow-up study, we may consider restricting the selection of study subjects for possible confounding factors to obtain homogeneous study subjects as much as possible. Still, the representativeness of the study subjects may be affected to some extent, in addition to using randomization to select study subjects to reduce confounding bias as much as possible.

## 5. Conclusions

Anxiety and depression significantly mediate the impact of childhood trauma and poor family functioning on IA in adolescents. Traumatic childhood experiences increase the risk of developing anxiety and depression in adolescents. Furthermore, anxiety and depression were significantly correlated with IA, and childhood trauma was positively associated with adolescent IA. Poor family functioning is a direct risk predictor of IA in adolescents. In addition, poor family functioning enhances the risk prediction of IA in adolescents by significantly increasing the likelihood of suffering from anxiety and depression. This conclusion provides the basis and direction for developing prevention strategies, interventions, and treatment measures for adolescent anxiety, depression, and IA from childhood. These developments should be aimed at improving the family environment in which adolescents grow up, reducing possible childhood traumas, and improving adolescents’ knowledge of the family concept, thereby reducing the prevalence of adolescents’ negative emotions and IA.

## Figures and Tables

**Figure 1 ijerph-19-13639-f001:**
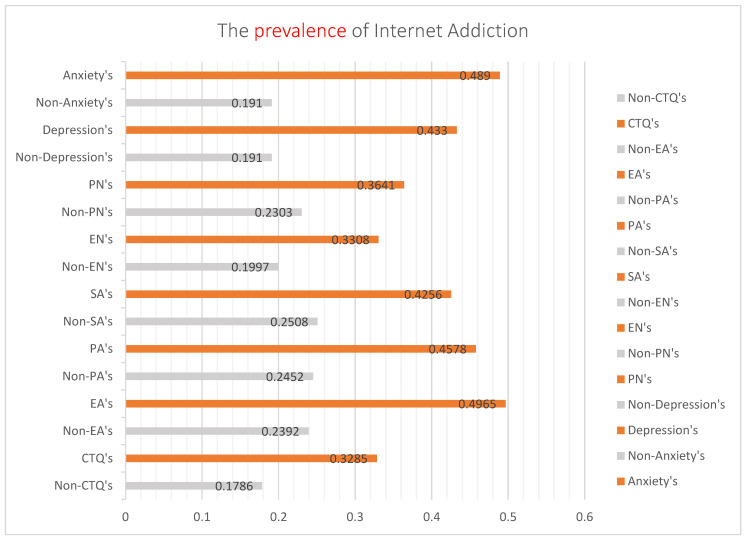
Cluster bar graph analysis of the prevalence of Internet addiction among adolescents with or without childhood trauma, anxiety, or depression. Note: Non-CTQ’s = None childhood trauma. CTQ’s = With childhood trauma. Non-EA’s = None emotion abuse. EA’s = With emotion abuse. Non-PA’s = None physical abuse. PA’s = With physical abuse. Non-SA’s = None Sexual abuse. SA’s = With Sexual abuse. Non-EN’s = None emotion neglect. EN’s = With emotion neglect. Non-PN’s = None physical neglect. PN’s = With physical neglect.

**Figure 2 ijerph-19-13639-f002:**
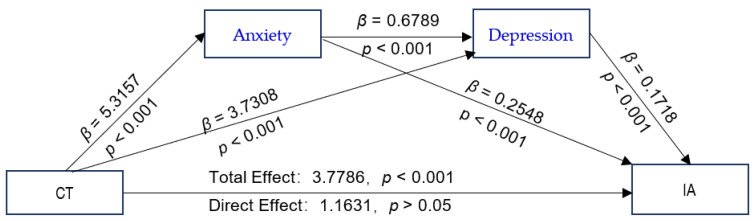
The chain-mediated model predicting adolescents’ IA based on the experience of childhood trauma mediated by anxiety and depression. Abbreviations: CT, childhood trauma; IA, Internet addiction.

**Figure 3 ijerph-19-13639-f003:**
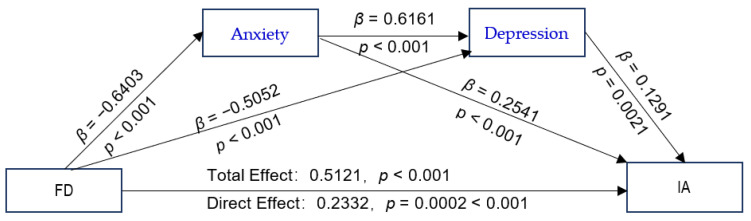
The chain-mediated model predicting adolescents’ Internet addiction based on family dysfunction mediated by anxiety and depression. Abbreviations: FD, family dysfunction; IA, Internet addiction.

**Table 1 ijerph-19-13639-t001:** Comparison of the differences in the prevalence of Internet addiction between groups.

Characteristics (Code)	n (N = 3357, %)	IA (n, %)	χ^2^	ν	*p*-Value	r
Sex	3323 (99.0)	870 (26.2)	0.105	1	0.746	0.006
(1) Male	1681 (50.1)	436 (25.9)				
(2) Female	1642 (49.4)	434 (26.4)				
Single child	3311 (98.6)		4.837	1	0.028	−0.038 *
(1) yes	2315 (69.9)	631 (27.3)				
(2) no	996 (30.1)	235 (23.6)				
Grade	3357 (100)		207.950	6	0.000	0.228 **
6th Grade	1022 (30.4)	148 (14.5)				
7th Grade	598 (17.8)	139 (23.2)				
8th Grade	456 (13.6)	95 (20.8)				
9th Grade	128 (3.8)	37 (28.9)				
10th Grade	760 (22.6)	309 (40.7)				
11th Grade	187 (5.6)	53 (28.3)				
12th Grade	206 (6.1)	95 (46.1)				
Living Style	3219 (95.9)		5.665	3	0.129	0.031
(1) with parents	2605 (80.9)	694 (26.6)				
(2) with grandparents	232 (7.2)	67 (28.9)				
(3) with parents and grandparents	207 (6.4)	43 (20.8)				
(4) others	175 (5.4)	39 (22.3)				
Parental bias	2759 (82.2)		14.684	2	0.001	0.072 **
(1) no	2256 (81.8)	577 (25.6)				
(2) for Participants	345 (12.5)	113 (32.8)				
(3) for siblings	158 (5.7)	57 (36.1)				
Education level of the father	2208 (65.8)		11.601	3	0.009	−0.059 **
(1) 0–9 years	350 (15.9)	114 (32.6)				
(2) 10–12 years	485 (22.0)	134 (27.6)				
(3) 13–17 years	1092 (49.5)	258(23.6)				
(4) over 17 years	281 (12.7)	73 (26.0)				
Education level of the mother	2191 (65.3)		4.852	3	0.183	−0.042 *
(1) 0–9 years	407 (18.6)	118 (29.0)				
(2) 10–12 years	450 (20.5)	119 (26.4)				
(3) 13–17 years	1123 (51.3)	290 (25.8)				
(4) over 17 years	211 (9.6)	44 (20.9)				
Parental Marriage Quality	3152 (93.9)		48.659	4	0.000	0.084 **
(1) Excellent	2323 (69.2)	535 (23.0)				
(2) Good	428 (12.7)	151 (35.3)				
(3) Conflicted	105 (3.1)	44 (41.9)				
(4) Living apart	58 (1.7)	19 (32.8)				
(5) Divorce	238 (7.1)	75 (31.5)				
Family income	2493 (74.3)		2.546	3	0.467	0.001
(1) 0–100 K	636 (25.5)	150 (23.6)				
(2) 100–300 K	1120 (44.9)	299 (26.7)				
(3) 300–500 K	447 (17.9)	108 (24.2)				
(4) More than 500 K	290 (11.6)	71 (24.5)				
Satisfaction with household economy						−0.236 **
Self-evaluation of network usage troubles	3265 (97.3)		287.629	2	0.000	
(1) no	1939 (59.4)	308 (15.9)				
(2) yes	472 (14.5)	232 (49.2)				
(3) other Stresses	854 (26.2)	317 (37.1)				
Child Trauma	3357 (100)		96.801	1	0.000	
(1) no	1512 (45.0)	270 (17.9)				
(2) yes	1845 (55.0)	606 (32.8)				
EA	3357 (100)		89.244	1	0.000	
(1) no	3073 (91.5)	735 (23.9)				
(2) yes	284 (8.5)	141 (49.6)				
PA	3357 (100)		54.058	1	0.000	
(1) no	3108 (92.6)	762 (24.5)				
(2) yes	249 (7.4)	114 (45.8)				
SA	3357 (100)		29.117	1	0.000	
(1) no	3162 (94.2)	793 (25.1)				
(2) yes	195 (5.8)	83 (42.6)				
EN	3357 (100)		74.498	1	0.000	
(1) no	1788 (53.3)	357 (20.0)				
(2) yes	1569 (46.7)	519 (33.1)				
PN	3357 (100)		55.046	1	0.000	
(1) no	2588 (77.1)	596 (23.0)				
(2) yes	769 (22.9)	280 (36.4)				
Depression	3350 (99.8)		209.039	1	0.000	
(1) no	2375 (70.9)	454 (19.1)				
(2) yes	975 (29.1)	422 (43.3)				
Anxiety	3350 (99.8)		279.150	1	0.000	
(1) no	2557 (76.3)	488 (19.1)				
(2) yes	793 (23.7)	388 (48.9)				

Abbreviations: EA, emotional abuse; PA, physical abuse; SA, sexual abuse; EN, emotional neglect; PN, physical neglect ** p* < 0.05, *** p* < 0.01.

**Table 2 ijerph-19-13639-t002:** Correlation analysis between the total scores of IAT-20 and SAS, SDS, CTQ-28, SSFD, and FAD.

	IAT-20	SAS	SDS	EA	PA	SA	EN	PN	FA	IN	SL	IC	SSFD	BC
IAT-20														
SAS	0.403 **													
SDS	0.383 **	0.708 **												
EA	0.312 **	0.442 **	0.454 **											
PA	0.168 **	0.269 **	0.256 **	0.544 **										
SA	0.111 **	0.177 **	0.126 **	0.289 **	0.394 **									
EN	0.159 **	0.354 **	0.488 **	0.405 **	0.315 **	0.107 **								
PN	0.167 **	.0322 **	0.412 **	0.380 **	0.313 **	0.233 **	0.508 **							
FA	−0.297 **	−0.427 **	−0.545 **	−0.424 **	−0.292 **	−0.086 **	−0.584 **	−0.376 **						
IN	−0.205 **	−0.316 **	−0.433 **	−0.346 **	−0.280 **	−0.093 **	−0.433 **	−0.301 **	0.703 **					
SL	−0.170 **	−0.183 **	−0.169 **	−0.201 **	−0.085 **	−0.065 **	−0.120 **	−0.146 **	0.024	−0.042*				
IC	−0.009	−0.090 **	−0.197 **	−0.093 **	−0.116 **	−0.032	−0.232 **	−0.178 **	0.416 **	0.434 **	−0.285 **			
SSFD	−0.138 **	−0.253 **	−0.387 **	−0.260 **	−0.228 **	−0.059 **	−0.430 **	−0.273 **	0.793 **	0.825 **	−0.405 **	0.720 **		
BC	0.207 **	0.244 **	0.327 **	0.165 **	0.100 **	0.073 **	0.240 **	0.187 **	−0.375 **	−0.233 **	−0.121 **	−0.149 **	−0.254 **	
GF	0.311 **	0.469 **	0.585 **	0.482 **	0.306 **	0.078 **	0.568 **	0.409 **	−0.723 **	−0.551 **	−0.254 **	−0.236 **	−0.493 **	0.455 **

Abbreviations: IAT, Internet Addiction Test; SAS, self-rating anxiety scale; SDS, self-rating depression scale; EA, emotional abuse; PA, physical abuse; SA, sexual abuse; EN, emotional neglect; PN, physical neglect; FA, family atmosphere; IN, individualized; SL, systematic logic; IC, illness concept; SSFD, Self-rating Scale of Systematic Family Dynamics, revised version; BC, behavioral control; GF, general function subscale; FAD, family assessment device; CTQ-28, Childhood Trauma Questionnaire. Note: ** *p* < 0.01.

**Table 3 ijerph-19-13639-t003:** The chain-mediated effects of childhood trauma on Internet addiction and bias-corrected 95% confidence intervals.

Effect	Effect Model	Coeff	95% CI [LLCI, ULCI]	Effect Ratio%
Direct Effect	CTQ→IAD	1.1631	[−0.2436, 2.5697]	
Indirect Effect	Ind1	1.3544	[0.8085, 1.9752]	52.3
	Ind2	0.6410	[0.2556, 1.0421]	24.1
	Ind3	0.6201	[0.2571, 0.9930]	23.6
	Ind1 minus Ind2	0.7134	[−0.0842, 1.6199]	
	Ind1 minus Ind3	0.7343	[−0.0387, 1.5964]	
	Ind2 minus Ind3	0.0210	[−0.1859, 0.2335]	
Total Indirect Effect		2.6155	[1.9555, 3.3105]	100
Total Effect		3.7786	[2.3780, 5.1792]	100

Note. Indirect effect key: Ind1: CTQ→SAS→IAD; Ind2: CTQ→SDS→IAD; Ind3: CTQ→SAS→SDS→IAD.

**Table 4 ijerph-19-13639-t004:** The chain-mediated effects of family dysfunction on Internet addiction and bias-corrected 95% confidence intervals.

Effect	Effect Model	Coeff	95% CI [LLCI, ULCI]	Effect Ratio%
Direct Effect	GF→IAD	0.2332	[0.1099, 0.3565]	45.5
Indirect Effect	Ind1	0.1627	[0.0997, 0.2325]	31.8
	Ind2	0.0652	[0.0141, 0.1193]	12.7
	Ind3	0.0509	[0.0114, 0.0902]	10.0
	Ind1 minus Ind2	0.0975	[−0.0073, 0.2064]	
	Ind1 minus Ind3	0.1118	[0.0231, 0.2089]	
	Ind2 minus Ind3	0.0143	[−0.0011, 0.0376]	
Total Indirect Effect		0.2789	[0.2068, 0.3544]	54.5
Total Effect		0.5121	[0.4001, 0.6241]	100

Note. Indirect effect key: Ind1: GF→SAS→IAD; Ind2: GF→SDS→IAD; Ind3: GF→SAS→SDS→IAD.

## Data Availability

Based on informed consent, the data involved in this study are for use in this project only.

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
