# Peer review of "The Influence of Childhood Trauma and Family Functioning on Internet Addiction in Adolescents: A Chain-Mediated Model Analysis"

_ijerph, 2022, doi:10.3390/ijerph192013639_

Round 1

Reviewer 1 Report (Previous Reviewer 3)

Thanks for addressing my comments and concerns appropriately. I am fine with this R1. 

Author Response

Responses: Thank you for your careful examination. Based on your comments, we have thoroughly checked and tried to correct grammatical and spelling errors in the manuscript with the help of Grammarly Business Edition software.

Reviewer 2 Report (New Reviewer)

Introduction

-Authors have not neither commented nor explored the option that internet addiction (i.e., heavy use) during adolescence come because of an early exposure to the internet during childhood. There is some research conducted on this, and I think it is worth mentioning in the introduction.

Methods.

-Could the authors kindly provide the power and the alfa for determining the sample size.

-Authors should better explain how they conducted regression analysis of the chain mediation model.

Results

-In table 2, the variables row should also present abbreviations for variables.

-In Figures 2 and 3, how do you estimate the direct and total effect of the exposure over the variable if you are not using multivariable regression modes?

Discussion

-The first paragraph of the discussion should start by summarizing your main findings as well as highlighting what you are adding to the current state of the art.

-In the limitations, since you are not adjusting your analyses, there is also a chance for a residual confounding bias.

Author Response

Dear reviewer,  Thank you very much for your valuable comments and suggestions. we had provided a point-by-point response as following.

  1. English language and style are fine/minor spell check required

Responses: Thank you for your careful examination. Based on your comments, we have thoroughly checked and tried to correct grammatical and spelling errors in the manuscript with the help of Grammarly Business Edition software.

Introduction

  1. Authors have not neither commented nor explored the option that internet addiction (i.e., heavy use) during adolescence come because of an early exposure to the internet during childhood. There is some research conducted on this, and I think it is worth mentioning in the introduction.

Responses: Thank you very much for your valuable comments. Based on your suggestions, I rechecked the latest literature and summarized these aspects in the introduction, as detailed in lines 47-51, and added relevant references in the references.

Methods.

  1. Could the authors kindly provide the power and the alfa for determining the sample size.

Responses: We are grateful for your careful review. The sample size of the study was estimated considering the pre-survey information and general professional requirements, and the prevalence of IA in adolescents was about 23.4% in the pretest of 189 samples. 6% relative error is allowed in the study. α is the size of a class of errors and the choice of a bilateral cut-off value is appropriate.

  1. Authors should better explain how they conducted regression analysis of the chain mediation model.

Responses: Thank you for your professional opinion. The regression analysis of the chain mediation model was conducted considering the possible interactions between mediating variables, but this study only explored and analyzed the most common ones, and based on your suggestion, we will explore more multiple mediation analysis for the existence of effects between mediating variables in subsequent studies.

Results

  1. In table 2, the variables row should also present abbreviations for variables.

Responses: Thank you for your suggestion, the abbreviations of the variables have been listed in the variable row of Table 2 based on your suggestion.

  1. In Figures 2 and 3, how do you estimate the direct and total effect of the exposure over the variable if you are not using multivariable regression modes?

Responses: Thank you very much for your specific comments, this study explored some of the most important predictor and predicted variables in univariate analysis, considering the relevant mediating variables, and in the future, regression models for multiple variables combined with mediating models for structural model equations could be considered, so as to estimate the direct and total effects of exposure on the variables.

Discussion

  1. The first paragraph of the discussion should start by summarizing your main findings as well as highlighting what you are adding to the current state of the art.

Responses: we sincerely thank you for your valuable comments, and based on your suggestions, we have provided a brief summary of the main findings of this study in the first paragraph of the discussion, in lines 314-327.

  1. In the limitations, since you are not adjusting your analyses, there is also a chance for a residual confounding bias.

Responses: We fully agree with your comments and suggestions, and according to your suggestions, we have added limitations to the manuscript. In the follow-up study, we may consider restricting the selection of study subjects for possible confounding factors to obtain homogeneous study subjects as much as possible, but the representativeness of the study subjects may be affected to some extent, in addition to using randomization to select study subjects so as to reduce confounding bias as much as possible. As detailed in lines 369-375.

Round 2

Reviewer 2 Report (New Reviewer)

The authors addressed well my comments. 

This manuscript is a resubmission of an earlier submission. The following is a list of the peer review reports and author responses from that submission.

Round 1

Reviewer 2 Report

Thank you for giving me the opportunity to review this manuscript. This study examined the china mediation of anxiety and depression in the effects of childhood trauma and family functioning on adolescent Internet addiction.  

1. Please check the grammar and spelling. There are many tense and grammatical mistakes that affect the readers’ understanding.

2. The authors need to check the use of statistical terms and obey the fundamental norm. For example, when you provide the value of χ2, please give the readers the value of degree of freedom.

3. This manuscript lacks the important background that the authors propose the hypotheses. In the INTRODUCTION, the authors need to find a theoretical framework to explain why anxiety and depression have a china mediating effect on the relationship of childhood trauma and family functioning with adolescent Internet addiction.

4. Please add the analytic strategies in the section of Methods.

5. What is the relationship between childhood trauma with poor family functioning? Why did the authors decide to analyze the chain mediating role of anxiety and depression in the effect of childhood trauma and poor family functioning on Internet addiction, respectively?

6. Please provide reliability and validity of the scales used in this manuscript.

7. The authors need to reconsider how to discuss these findings. What are these findings consistent or inconsistent with previous results? What did these results tell us and how to explain our findings?

8. The authors need to emphasize what new knowledge this manuscript contributes to. 

Good luck.

Reviewer 3 Report

Thanks for inviting me to review this research work which has explored the current situation of Internet addiction among Chinese adolescents in Shanghai. In addition, the authors have established a chain mediation model using the data to demonstrate the chain mediated of anxiety and depression. The sample size was adequate, but the sampling approach was neither clearly presented nor well justified. This is the first major problem. Second, the literature review is missing. Without a good literature review, you cannot justify your hypotheses for this study. Third, you need to have a leading research question in addition to the four hypotheses. Fourth, you need to report the reliability and validity of each measure used in this study. Without a reliable measure, how could you guarantee that you are measuring the target variables? Last but not least, the writing quality is far below the basic requirements.  Please have it rewritten by a professional English editor.